# Broad-spectrum activity against mosquito-borne flaviviruses achieved by a targeted protein degradation mechanism

Han-Yuan Liu ©[1,6], Zhengnian Li[2,6], Theresia Reindl ©[1], Zhixiang He[3], Xueer Qiu[4], Ryan P. Golden ©[2], Katherine A. Donovan ©[3,5], Adam Bailey[4], Eric S. Fischer ©[3,5], Tinghu Zhang[2], Nathanael S. Gray ©[2] ✉ & Priscilla L. Yang ©[1] ✉

Viral genetic diversity presents significant challenges in developing antivirals with broad-spectrum activity and high barriers to resistance. Here we report development of proteolysis targeting chimeras (PROTACs) targeting the dengue virus envelope (E) protein through coupling of known E fusion inhibitors to ligands of the CRL4[CRBN] E3 ubiquitin ligase. The resulting small molecules block viral entry through inhibition of E-mediated membrane fusion and interfere with viral particle production by depleting intracellular E in infected Huh 7.5 cells. This activity is retained in the presence of point mutations previously shown to confer partial resistance to the parental inhibitors due to decreased inhibitor-binding. The E PROTACs also exhibit broadened spectrum of activity compared to the parental E inhibitors against a panel of mosquito-borne flaviviruses. These findings encourage further exploration of targeted protein degradation as a differentiated and potentially advantageous modality for development of broad-spectrum direct-acting antivirals.

Most approved antivirals in use today are small molecules that bind to and interfere with the function of essential viral proteins, in most cases viral enzymes (*e.g.*, viral polymerases, proteases). These direct-acting antivirals (DAAs) can be highly effective when used in combination, providing a cure for hepatitis C virus (HCV) and allowing long-term control over human immunodeficiency virus (HIV). Despite these successes, DAA inhibitors generally suffer from narrow spectrum activity ("one bug, one drug"), and resource limitations pose significant challenges for the development of equivalent DAAs against other viral pathogens of biomedical concern. The ability of RNA viruses to generate genetic diversity is also a major driver of antiviral drug resistance because naturally resistant variants may preexist and emerge upon initiation of treatment, as has been observed with the use of individual DAAs as monotherapies against HIV[1], HCV[2], and influenza virus[3,4], and because mutations that reduce drug susceptibility can be

generated by low levels of replication during antiviral monotherapy treatment. Systematic approaches to increase potency and broaden efficacy across related species of viruses are needed.

Targeted protein degradation (TPD) has emerged as a pharmacological modality in which small molecules ("degrader molecules") induce elimination of the target protein through cellular protein degradation pathways. The most commonly used degrader molecules are called proteolysis targeting chimeras (PROTACs) and feature two ligands, one for the target of interest and the other for an E3 ubiquitin ligase connected via a linker. Functionally, they simultaneously bind to the target and the E3 ligase, thus inducing proximity that results in ubiquitination of the target protein and its subsequent degradation by the proteasome. This unique mechanism of action has afforded some key advantages over target inhibition in the development of anticancer agents, namely improved selectivity[5,6] and potency[5,7,8], as well as the

---

[1]Department of Microbiology and Immunology, Stanford University School of Medicine, Stanford, CA, USA. [2]Department of Chemical and Systems Biology, Chem-H and Stanford Cancer Institute, Stanford University School of Medicine, Stanford, CA, USA. [3]Department of Cancer Biology, Dana-Farber Cancer Institute, Boston, MA, USA. [4]Department of Pathology & Laboratory Medicine, University of Wisconsin–Madison, Madison, WI, USA. [5]Department of Biological Chemistry and Molecular Pharmacology, Harvard Medical School, Boston, MA, USA. [6]These authors contributed equally: Han-Yuan Liu, Zhengnian Li. ✉e-mail: nsgray01@stanford.edu; ply@stanford.edu

potential to overcome the effects of drug resistance-causing mutations[9–12]. The application of TPD as an antiviral strategy, however, has been limited[13–15]. We previously showed that we could repurpose telaprevir, a small molecule inhibitor of the HCV NS3-4A protease, by conjugating it via a linker to a ligand of cereblon (CRBN), the substrate recognition subunit of the CUL4A/B-RBX1-DDB1-CRBN (CRL4$^{CRBN}$) E3 ubiquitin ligase[13]. The resulting PROTACs potently inhibited HCV in cell culture due to induced degradation of NS3 as well as inhibition of NS3-4A activity, and this antiviral activity was maintained even in the presence of point mutations that confer partial resistance to telaprevir. While in this case, conversion of telaprevir to a PROTAC resulted in a DAA with improved resilience to point mutations with decreased drug-binding, the generalizability of this approach to other DAAs and whether this aspect of TPD can be leveraged to broaden the activity spectrum has yet to be established. Other groups have since reported development of PROTACs to target the neuraminidase[15] and hemagglutinin proteins[16] of influenza virus as well as the serendipitous discovery of a microbial metabolite that induces degradation of the influenza virus PA endonuclease protein[17]. Together, these published studies demonstrate the feasibility of TPD as an antiviral approach and provide motivation to investigate TPD as a method to improve resistance profiles and/or broaden the spectrum of activity of DAAs.

Flaviviruses are a genus of positive-stranded RNA viruses spread by arthropod vectors. Dengue virus (DENV) is currently the most widespread, with more than 400 million people infected annually. Other mosquito-borne members of the genus causing disease in humans include Zika (ZIKV), Japanese encephalitis (JEV), West Nile (WNV), and yellow fever viruses (YFV). Due to climate change and globalization, these viruses are projected to continue to spread. Despite being a major global health threat, effective DAAs and vaccine-based strategies against flaviviruses remain major, unmet biomedical needs. Since the development of DAAs against all individual members of this group of viruses is resource-limited, the availability of broad-spectrum DAAs that are effective against multiple flavivirus species would represent a significant breakthrough for the field.

The flavivirus envelope (E) protein exists as 90 prefusion dimers on the surface of the virion and is essential for multiple steps in the replication cycle of the virus. First, it mediates the initial attachment step of viral entry through interaction of its domain III with host factors on the extracellular face of the plasma membrane. Following internalization of the virion by clathrin-dependent endocytosis, acidification of the endosomal compartment triggers reorganization and refolding of E as a postfusion trimer. These significant structural changes provide the driving force for fusion of the viral lipid bilayer with the endosomal membrane, creating a fusion pore that allows the viral genome to escape to the cytoplasm where it can be expressed. The E protein is also essential at later stages of the replication cycle, in that no new infectious virions can be produced in the absence of E. Mutagenesis studies have identified point mutations with effects on entry or viral particle production as well as mutations that affect both processes[18–23], suggesting that small molecules targeting E might act through one or more modes of action. The development of small molecule inhibitors of DENV E-mediated fusion was facilitated by serendipitous co-crystallization of the detergent n-octyl-β-D-glucoside (βOG) in a conserved pocket (dubbed the "βOG pocket") located between domains I and II[24] (Fig. 1A), wherein residues responsible for altering the pH threshold for the fusion of related flaviviruses had been previously identified[25–30]. The availability of this high-resolution co-crystal structure as well as assays to monitor E's biochemical function in membrane fusion enabled the discovery and optimization of multiple series of small molecules that bind to prefusion DENV E and that exert antiviral activity by inhibiting fusion during viral entry[31–35].

The path to develop small molecules that inhibit virion production by binding to E has been less obvious. On newly formed, immature virions, E associates with its viral chaperone, prM, and exists as 60 trimeric spikes, each spike containing three prM-E heterodimers[36,37] ((prM/E)$_3$). Differences in the positions of domains I and II on mature and immature virions significantly affect the βOG pocket (Fig. 1B). As the immature virions traffic through the secretory system, they undergo a maturation process that culminates in release of mature virions bearing the characteristic 90 prefusion dimers of E in the extracellular space. While these structural changes have been well-characterized[36–38], ways to interfere with this process using small molecules have not been identified. In addition, the ability of anti-prM antibodies to mediate infection of completely immature viral particles suggests that blocking E's structural changes during virion maturation might not have the desired antiviral effect[39]. An alternate approach might be to target E before new particles form; however, E's structure prior to virion budding and whether interaction with a small molecule could inhibit the formation of virions are not known.

Here, we describe characterization of PROTACs that induce proteosome-mediated degradation of DENV E. This activity is associated with significantly increased antiviral potency compared to the parental E fusion inhibitors from which they were derived. We show that this increased antiviral potency is dependent on the TPD mechanism and that the E degraders have a dual mode of action, acting as inhibitors blocking fusion during viral entry while also affecting viral particle production by inducing CRBN-dependent degradation of intracellular E protein. The activity against viral particle production is maintained in a virus-like particle system in the presence of point mutations that were previously shown to confer partial resistance to the parental E fusion inhibitors due to reduced binding. Further, we show that the E degraders exhibit significantly increased antiviral activity against related mosquito-borne flaviviruses including ZIKV, JEV, WNV, and YFV, when compared to the parental inhibitors.

## Results
### Design of degraders to target the dengue E protein
To design PROTACs against the DENV E protein, we selected well-characterized E fusion inhibitors from two structurally independent lead series as E-targeting ligands. GNF-2, a 4,6-disubstituted pyrimidine, and 2-12-2, a 2,4-diamino substituted pyrimidine (Fig. 1C), were previously shown to bind to prefusion E on the surface of mature DENV virions and to block infection by inhibiting E-mediated membrane fusion[31,40]. Although high-resolution structures of these molecules bound to the DENV E protein have not been solved, we used insights from prior site-directed mutagenesis and medicinal chemistry studies[31,40] to inform our PROTAC design and attach the linker in a way that avoids deleterious effects on the interaction with E and that should allow for effective E3 ligase recruitment. To recruit an E3 ligase, we chose ligands of CRBN, the substrate recognition subunit of the CRL4$^{CRBN}$ E3 ubiquitin ligase. The medicinal chemistry effort required to develop these PROTACs, including DC$_{50}$ and DC$_{max}$ determination and proteomic evaluation of degrader specificity are being reported separately[41]. These efforts led to the identification of two degrader candidates, ZXH-2-107 and ZXH-8-004, derived from GNF-2 and 2-12-2, respectively, and which will hereafter be referred to as "GNF-2-deg" and "2-12-2-deg" for simplicity (Fig. 1D). To distinguish between the inhibition- and degradation-dependent pharmacology of GNF-2-deg and 2-12-2-deg, we synthesized the corresponding negative control compounds, GNF-2-deg-BUMP, and 2-12-2-deg-BUMP (Fig. 1D), by replacing the glutarimide ring on thalidomide with a δ-lactam moiety to greatly reduce interaction with CRBN[41], which was confirmed experimentally[41].

### E degraders deplete intracellular E via a CRBN- and proteasome-dependent mechanism
To verify that GNF-2-deg and 2-12-2-deg function as E degraders in the context of viral infection, we first characterized their effects on the

intracellular abundance of E in DENV2-infected cells as outlined in Fig. 2A. GNF-2-deg (DC$_{50}$ 0.83 µM, DC$_{max}$ 99%) and 2-12-2-deg (DC$_{50}$ 0.21 µM, D$_{max}$ 95%)[41] both cause a concentration-dependent reduction of intracellular E protein. This effect of E is CRBN-dependent, as it is not observed with negative control compounds GNF-2-deg-BUMP and 2-12-2-deg-BUMP or when the infection is performed in CRBN-deficient cells (CRBN$^{-/-}$) (Fig. 2B and Supplementary Fig. 1). Likewise, the depletion of E by GNF-2-deg and 2-12-2-deg is not observed in the presence of neddylation inhibitor MLN4924 or proteasome inhibitor MG-132 or when excess lenalidomide is present to compete with the E degraders for binding to CRBN (Fig. 2C–F). Together, these

experiments support the interpretation that GNF-2-deg and 2-12-2-deg act as bona fide degraders of DENV E in virus-infected cells.

## GNF-2-deg and 2-12-2-deg exhibit increased antiviral potency compared to the parental E inhibitors

To determine if depletion of E by GNF-2-deg and 2-12-2-deg results in significant antiviral activity, we examined their effects on the replication of live DENV2 in cell culture as outlined in Fig. 2A. The two E degraders caused significant concentration-dependent reductions in viral yield in the absence of appreciable cytotoxicity (Fig. 3A and Supplementary Fig. 2). Interestingly, GNF-2-deg and 2-12-2-deg exhib-

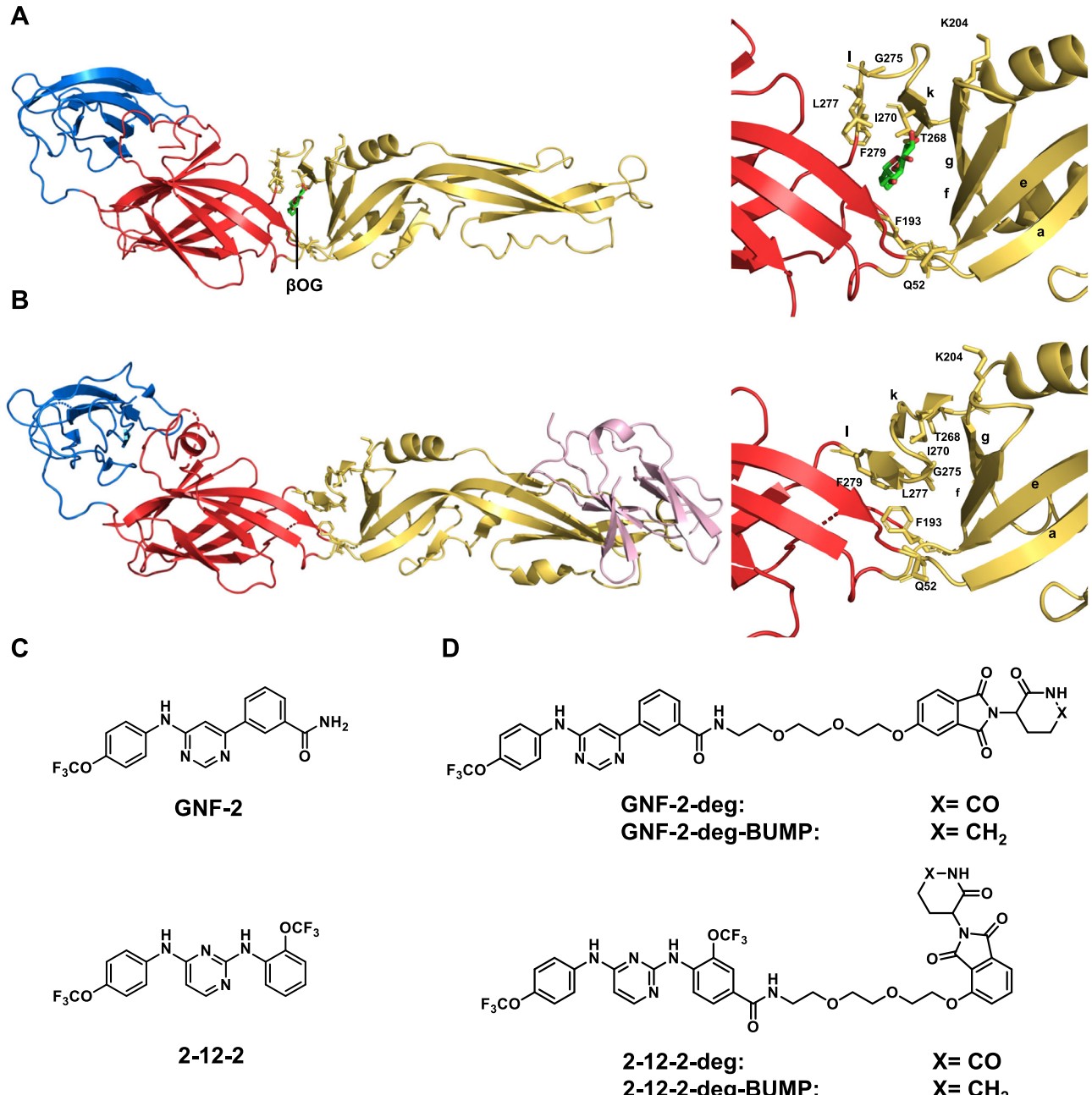

**Fig. 1 | Design of bivalent degraders targeting the DENV E protein. A** Structure of the soluble DENV2 prefusion E protein with βOG bound in the pocket located between domains I and II (PDB:1OKE). Note that only one monomer of the prefusion E is illustrated here for visual simplicity. **B** Structure of the DENV2 immature prM-E proteins (PDB:3C6E). Only one monomer of the immature trimer is shown. Domains I (red), II (yellow), III (blue), βOG (green), and prM (pink) are shown. **C** Chemical structures of GNF-2 and 2-12-2. **D** DENV E PROTACs GNF-2-deg, 2-12-2-deg, and their corresponding negative control compounds, GNF-2-deg-BUMP and 2-12-2-deg-BUMP, in which truncation of the carbonyl group at the glutarimide disrupts the CRBN-binding moiety.

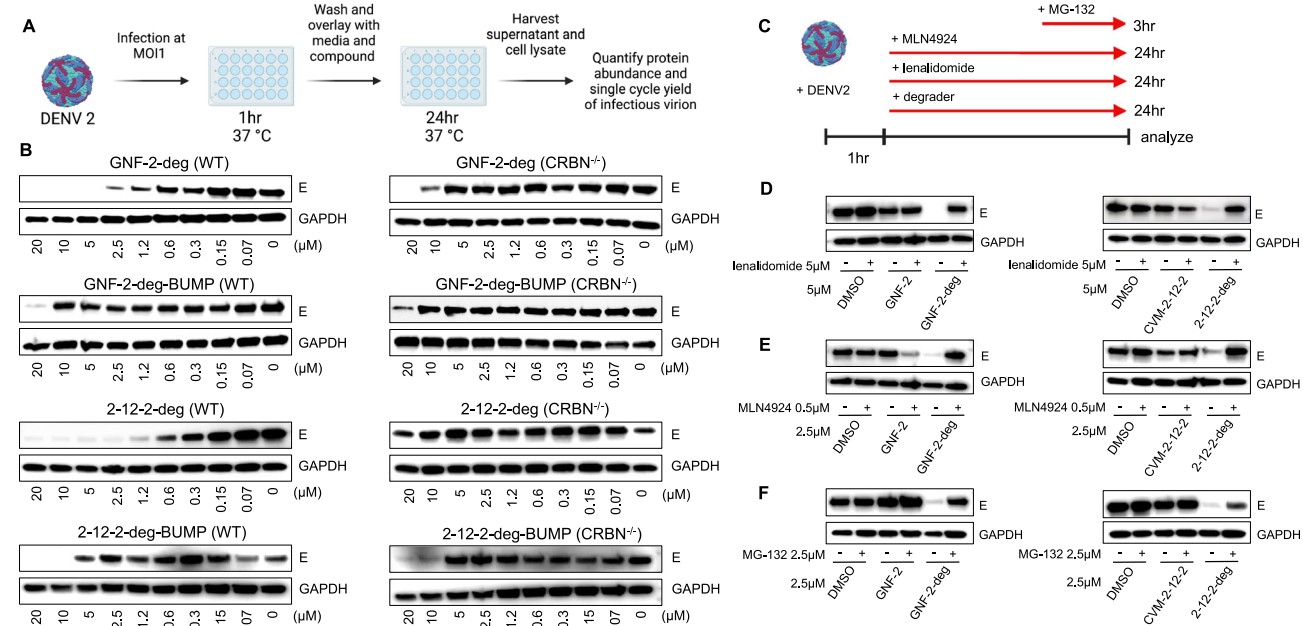

**Fig. 2 | GNF-2-deg and 2-12-2-deg induce CRBN- and proteasome-dependent depletion of DENV E in infected cells. A** Schematic of antiviral assay utilized in the study. **B** Western blots show concentration-dependent depletion of E in the presence of GNF-2-deg and 2-12-2-deg. Depletion of E is not observed for negative control compounds GNF-2-deg-BUMP or 2-12-2-deg-BUMP or when the experiment is conducted in CRBN-deficient cells. The representative results are shown from $n = 3$ independent experiments for GNF-2 derivative compounds in WT and $n = 2$ for all other conditions. **C** Schematic of experiments probing the mechanism of E degradation. Cells were infected with DENV2 at an MOI of 1 and treated with the E degrader (GNF-2-deg or 2-12-2-deg) along with the CRBN ligand lenalidomide or the neddylation inhibitor MLN4924 from 1 to 24 h post-infection. Due to cytotoxicity in the presence of proteasome inhibitor MG-132, cotreatment was limited to 21 to 24 h post-infection. Western blots show that depletion of E in the presence of GNF-2-deg and 2-12-2-deg is dependent on CRBN-binding (**D**), neddylation activity to activate CRL4$^{CRBN}$ (**E**), and proteasome activity (**F**). Representative results of lenalidomide treatment are shown from $n = 4$ and $n = 2$ independent experiments for GNF-2 and 2-12-2 derivatives, respectively. Representative results of MLN4924 treatment are shown from $n = 3$ and $n = 2$ independent experiments for GNF-2 and 2-12-2 derivatives, respectively. Representative results of MG-132 treatment are shown from $n = 2$ independent experiments. Figures 2A and 2C were created with Biorender.com.

ited significant increases in antiviral activity compared to their respective parental inhibitors, with GNF-2-deg exhibiting an antiviral EC$_{90 \text{ WT}}$ value of $3.50 \pm 1.52$ μM versus GNF-2's EC$_{90 \text{ WT}}$ value of $13.11 \pm 3.59$ μM and 2-12-2-deg having an EC$_{90 \text{ WT}}$ $1.67 \pm 0.71$ μM versus 2-12-2's EC$_{90 \text{ WT}}$ $13.27 \pm 0.05$ μM (Fig. 3A). This improvement in antiviral activity is CRBN-dependent, as it is not observed for GNF-2-deg-BUMP and 2-12-2-deg-BUMP or when GNF-2-deg and 2-12-2-deg are tested in CRBN$^{-/-}$ cells. Furthermore, we observed that DENV replication in the presence of the E degraders was at least partially rescued upon co-treatment with compounds to inhibit neddylation (MLN4924) or proteasome activity (MG-132) or to prevent CRL4$^{CRBN}$-engagement (excess lenalidomide), whereas these treatments had negligible impact on DENV replication in the presence of the parental E inhibitors (Fig. 3B–D). Taken together, the results provide strong evidence that GNF-2-deg and 2-12-2-deg utilize the CRL4$^{CRBN}$ E3 ligase to deplete intracellular DENV E protein and that this antiviral activity is associated with a notable enhancement of antiviral potency.

## GNF-2-deg and 2-12-2-deg have a dual mode of action: inhibition of E's function in viral entry and blocking of viral particle production due to a TPD mechanism

To examine the mode(s) of action of GNF-2-deg and 2-12-2-deg, we performed time of addition studies, first comparing antiviral activities of parental inhibitors and degraders in a previously described infectivity assay[31,32] in which compound treatment is limited to preincubation with the viral inoculum prior to infection. Under these conditions, GNF-2-deg and 2-12-2-deg exhibited antiviral activity comparable to that of parental inhibitors GNF-2 and 2-12-2 (Supplementary Fig. 3), consistent with the idea that these compounds bind to prefusion E on

the virion surface and inhibit E-mediated fusion during virus entry. Although we did not test explicitly whether targeted protein degradation contributes to antiviral activity in this assay, this seems unlikely since the ubiquitin-proteasome system is intracellular and inaccessible to both extracellular virions and the internalized virions still trapped in the endosome. Further making it unlikely that targeted protein degradation of E contributes to antiviral activity at this stage of the replication cycle, endosomal fusion of DENV in live-cells has been shown to occur between 5 to 17 min post-infection[42] and degradation of E once this step has occurred is unlikely to have an antiviral effect because the viral genome has been released. Rather, the antiviral activity exhibited in this assay most likely reflects the inhibitory activity of GNF-2 and 2-12-2, which is retained by their respective E degraders.

To examine the effects of GNF-2-deg and 2-12-2-deg on post-entry steps of the viral replication cycle, we conducted experiments in which compound treatment was delayed until after an initial one-hour infection period and the abundance of E in cell lysates and the yield of infectious viral particles in the culture supernatant were analyzed at 24 h post-infection. Both the parental inhibitors and the E degraders exhibited antiviral activity in this post-entry assay (Fig. 3A), suggesting that they can affect the post-entry stage of the viral replication cycle. Western blot analysis of core, NS4B, and NS5 in cell lysates indicated that these viral proteins were also depleted in the presence of GNF-2-deg and 2-12-2-deg at this time point and at the earlier time points at which we could reproducibly detect these proteins under these infection conditions (Supplementary Fig. 4). The reduced abundance of core, NS4B, and NS5 in these end-point measurements may reflect reduced expression of these proteins due to general antiviral activity and/or increased degradation of core, NS4B, and NS5 in the presence

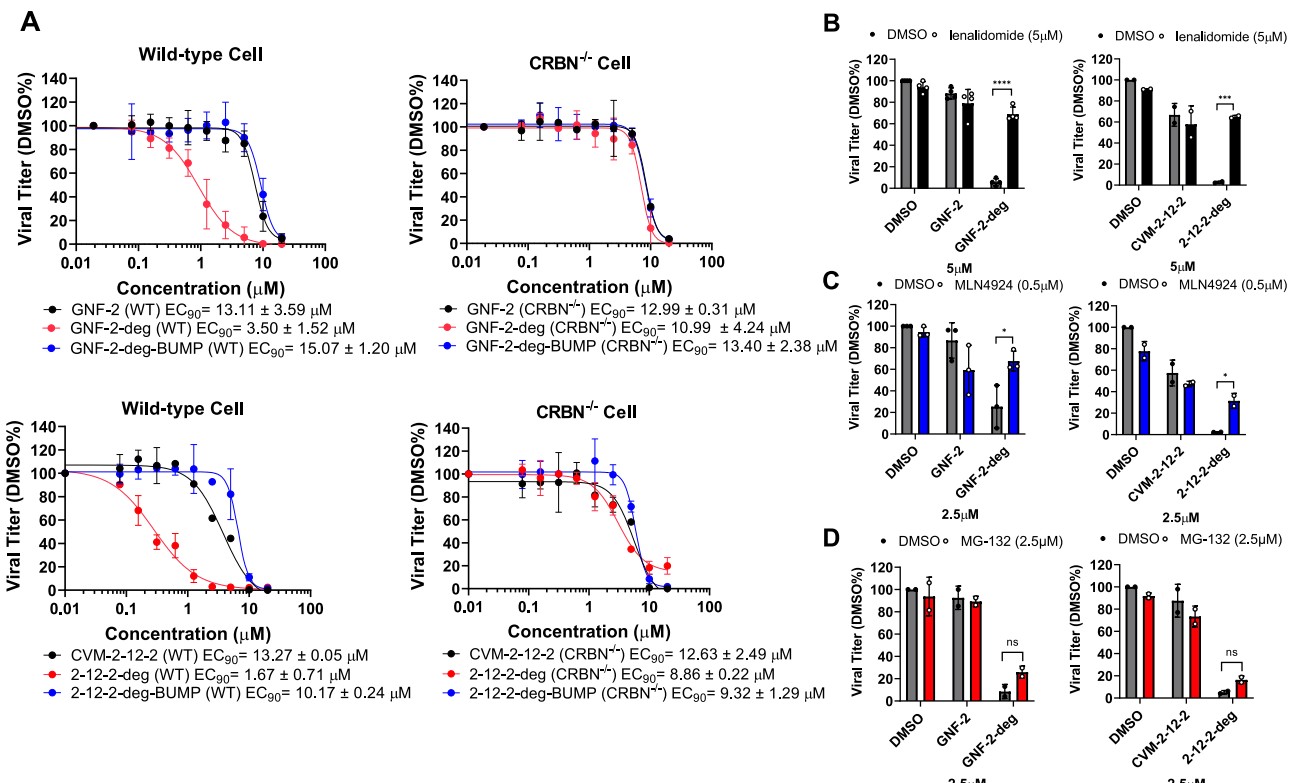

**Fig. 3 | GNF-2-deg and 2-12-2-deg have CRBN- and proteasome-dependent antiviral activity against DENV. A** Culture supernatants from the experiments depicted in Fig. 2 were harvested at 24 h post-infection to allow quantification of viral yield by plaque-formation assay. Antiviral $EC_{90}$ values were determined by nonlinear regression analysis of the data for both E degraders (GNF-2-deg and 2-12-2-deg) and the parental inhibitors. GNF-2-deg and 2-12-2-deg exhibit an increase in antiviral potency compared to GNF-2 and 2-12-2 that is CRBN-dependent. The representative results are shown from $n = 3$ independent experiments for GNF-2 derivative compounds in WT and $n = 2$ for all other conditions. Data are presented as means normalized to DMSO ± standard deviation. The antiviral activity of GNF-2-deg and 2-12-2-deg is blocked by the addition of excess lenalidomide ($p$ value < 0.0001 for GNF-2-deg and 0.0004 for 2-12-2-deg) (**B**), the addition of the neddylation inhibitor MLN4924 to block $CRL4^{CRBN}$ activity ($p$ value of 0.0287 for GNF-2-deg and 0.0274 for 2-12-2-deg) (**C**), or the addition of the proteasome inhibitor MG-132 ($p$ value of 0.0505 for GNF-2-deg and 0.0985 for 2-12-2-deg) (**D**). Representative results of lenalidomide treatment are shown from $n = 4$ and $n = 2$ independent experiments for GNF-2 and 2-12-2 derivatives, respectively. Representative results of MLN4924 treatment are shown from $n = 3$ and $n = 2$ independent experiments for GNF-2 and 2-12-2 derivatives, respectively. Representative results of MG-132 treatment are shown from $n = 2$ independent experiments. Data are presented as means normalized to DMSO ± standard deviation. Asterisks indicate that the differences between samples are statistically significant, using the unpaired two-tailed $t$ test. Data are presented as means normalized to DMSO ± standard deviation.

of GNF-2-deg and 2-12-2-deg, perhaps due to effects of the compounds on the viral polyprotein (Supplementary Note 1). With respect to mode of action, these observations opened the possibility that reduced viral gene expression and/or reduced RNA replication may contribute to the antiviral activity observed in these experiments.

To examine the effects of the compounds on viral particle production more directly, we used a virus-like particle (VLP) model system, which allows the study of DENV particle production in the absence of viral entry, polyprotein expression and processing, and genome replication[18,43,44]. Both the E inhibitors and the E degraders caused significant reductions in intracellular E and secreted VLPs in WT cells in these experiments when added at 4 h post-transfection with the prM-E expression plasmid (Fig. 4A). The effects of the fusion inhibitors GNF-2 and 2-12-2 on flavivirus particle production were not anticipated, as neither we nor others have previously reported this activity. While we do not yet know the mechanism(s) underlying the depletion of E in the presence of the inhibitors in this experimental model, it appears to differ from that of the E degraders. Notably, while the activities of GNF-2 and 2-12-2 were the same in WT and CRBN⁻/⁻ cells, the activities of GNF-2-deg and 2-12-2-deg were absent in CRBN⁻/⁻ cells. This is consistent with the idea that GNF-2-deg and 2-12-2-deg affect DENV viral particle production through a CRBN-dependent, TPD mechanism.

### The effect of E degraders on viral particle production is mediated by binding to E

Although parental inhibitors GNF-2 and 2-12-2 are validated ligands of DENV prefusion E, their binding site in the βOG pocket is not present on immature virions due to differences in the orientations of domains I and II on the surface of immature versus mature virions (Figs. 1B, 4B–C). CRBN-dependent depletion of intracellular E caused by GNF-2-deg and 2-12-2-deg suggests that the βOG pocket exists in the context of the polyprotein prior to cleavage or within E after it is cleaved from the polyprotein but prior to virion budding. Although it is also formally possible that the effect of GNF-2-deg and 2-12-2-deg in the VLP system is mediated by degradation or inhibition of an off-target and not due to binding of E, proteomics experiments to profile the degrader specificity of GNF-2-deg and 2-12-2-deg have not revealed any cellular targets expected to affect flavivirus particle production[41]. As a more explicit test of whether the activity of GNF-2-deg and 2-12-2-deg on VLP production is due to binding of E, we produced VLPs bearing mutations in the βOG pocket. The E-M196V mutation, which is located at the base of beta strand f, was identified when DENV2 was serially passaged in the presence of compound 7-148-6, an analog of 2-12-2[31]. This substitution was previously confirmed to confer partial resistance to 7-148-6 and a related analog of GNF-2[31,40] and to reduce binding of soluble prefusion E ($sE_2$) to 2-12-2 by >3-fold and to GNF-2 by >10-fold[31]. Substitutions

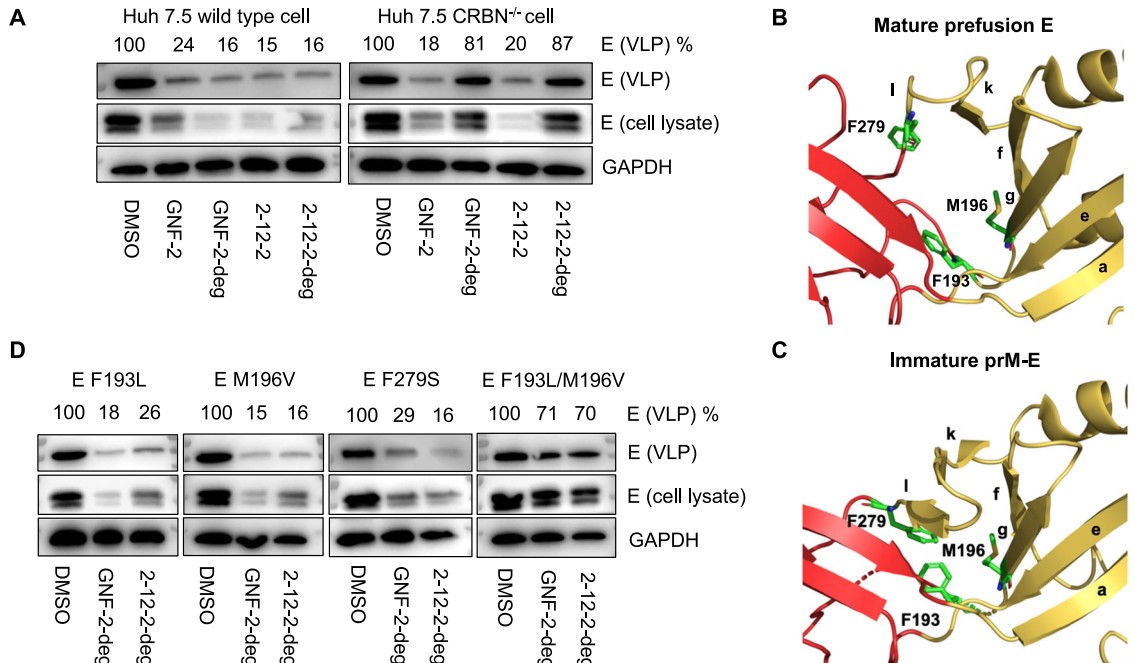

**Fig. 4 | GNF-2-deg and 2-12-2-deg inhibit viral particle production by a TPD mechanism.** Wild-type or CRBN⁻/⁻ Huh7.5 cells were transfected with the VLP expression plasmid, then treated with compounds at 4 h post-transfection. At 24 h post-transfection, culture supernatants and cell lysates were harvested to allow analysis of secreted VLPs and intracellular E. Representative results from $n \geq 2$ independent experiments are shown. **A** Both the degraders and parental inhibitors cause depletion of intracellular E associated with reduced VLPs in the culture supernatant; however, the activity of GNF-2-deg and 2-12-2-deg is lost in the CRBN⁻/⁻ cells. **B, C** Structures of the βOG pocket on soluble, mature, prefusion dimeric E

protein (PDB:1OKE) and prM-E protein on virions (PDB:3C6E), respectively, with domains I, II, III, and prM colored red, yellow, blue, and pink, respectively. Substitutions were introduced in the pocket at the hinge region at residues F193, M196, and F279 (green). **D** GNF-2-deg's and 2-12-2-deg's TPD-based inhibition of VLP formation is unaffected by single point mutations in the βOG pocket but is reduced when both F193L and M196V are introduced to the pocket. Cell lysates and purified VLPs were analyzed by Western blot to determine the effect of inhibitors and degraders on intracellular E abundance and VLP production. Representative results are shown from $n = 2$ independent experiments.

E-F193L and E-F279S were also previously characterized in a site-directed mutagenesis study and shown to reduce binding of sE₂ to 2-12-2, GNF-2, and related 2,4-diaminopyrimidine and 4,6-disubstitute pyrimidine E fusion inhibitors by 3- to >10-fold[31]. We examined whether these substitutions affect the activity of GNF-2-deg and 2-12-2-deg against intracellular E abundance and VLP production (Fig. 4D). At a single-concentration corresponding to the antiviral EC₉₀ value measured in the live virus assays, the effect of the E degraders on both intracellular E abundance and secreted VLPs was comparable for WT and the single mutation (E-F193L, E-M196V, and E-F279S) VLPs (Fig. 4A, D). Importantly, however, neither E degrader exhibited an effect on the E-F193L/M196V double mutant (Fig. 4D). This supports the idea that GNF-2-deg and 2-12-2-deg exert their effect on VLP production by binding to E and is consistent with the idea that the E PROTACs are less affected than the parental inhibitors to individual mutations that reduce compound-binding.

## E degraders exhibit broad-spectrum activity against multiple flaviviruses

In our previous characterization of GNF-2 and 2-12-2 as inhibitors of viral fusion during entry, we found that both compounds inhibit the entry of all four DENV serotypes (EC₉₀ values in the single-digit micromolar) with more modest activity against other mosquito-borne flaviviruses (JEV, WNV Kunjin)[31]. This prior work utilized infectivity assays in which antiviral activity was limited to the effect on E-mediated fusion during viral entry through washout of the compound following the initial one-hour infection period and/or through use of single-cycle reporter viruses that cannot produce new viral particles. To examine whether conversion of GNF-2 and 2-12-2 to the TPD-based mechanism of GNF-2-deg and 2-12-2-deg results in broader

spectrum activity, we compared the antiviral activities of the two parental inhibitors and the two degraders against ZIKV, JEV, WNV Kunjin, and YFV under conditions in which the compounds were present for the entirety of the viral life cycle to capture antiviral activity due to both modes of action (entry and particle production). The antiviral potencies of GNF-2 and 2-12-2 in these experiments are higher than what we previously observed[31,40], consistent with the observation that even the parental inhibitors can affect both entry and particle production; however, GNF-2-deg and 2-12-2-deg exhibit antiviral potencies greater than their respective parental inhibitors with antiviral EC₉₀ values consistently in the single-digit micromolar across ZIKV, JEV, and WNV Kunjin and with GNF-2-deg also having improved activity against YFV (Fig. 5). This gain of antiviral potency for the TPD-based antivirals echoes the CRBN-dependent improvement in antiviral potency that we observed against DENV (Fig. 3) and is correlated with reduced E protein in cell lysates for these other viruses (Supplementary Fig. 5). Of note, GNF-2 exhibits very weak activity against WNV Kunjin (EC₅₀ value > 20 μM) whereas GNF-2-deg has comparable activity (EC₉₀ values in the single-digit micromolar) against all four mosquito-borne flaviviruses in this panel. These data illustrate that conversion of an antiviral inhibitor to a degrader can be advantageous with respect to antiviral potency as well as spectrum of activity.

## Discussion

Although many viral proteins have multiple functions in the viral replication cycle, classical DAAs generally inhibit or derange only a single function of the viral enzyme being targeted. Pharmacological approaches that address the multifunctionality of a given viral protein might therefore enable higher antiviral potencies while also conferring an advantage in terms of avoiding drug resistance mediated by

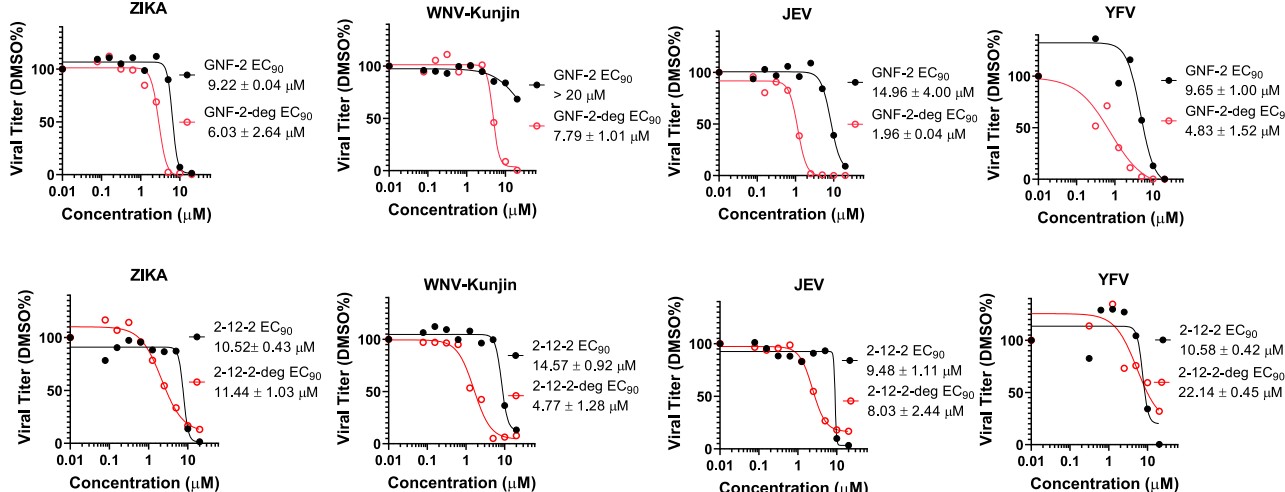

**Fig. 5 | E degraders exhibit greater antiviral potency and broader spectrum antiviral activity compared to parental E inhibitors.** Cells were infected with the tested virus at an MOI of 1, and infected cells were treated with the indicated compounds from 1 to 24 h post-infection, after which the yield of infectious virus in culture supernatants was quantified by plaque formation assay. Viral yields were normalized to those of the DMSO-treated controls. The antiviral EC$_{90}$, corresponding to the compound concentration resulting in a 90% reduction in viral titer, was determined by nonlinear regression. The data are presented as means of $n = 2$ independent experiments.

suppressor mutations that restore one but not all functions of the viral protein. One challenge in this approach is that it may not be possible to inhibit multiple functions of the viral protein by targeting a small molecule to a single site. An additional practical challenge is that many viral proteins have functions that, while proven essential in genetics experiments, are poorly understood in terms of biochemistry. This severely limits inhibitor development by conventional structure-based drug design and target-based screening approaches. Targeted protein degradation has the potential to mitigate some of these challenges. First, degradation of the viral protein removes all of its functions from the host cell, including functions that may not be known or well-characterized. Second, ligands of the viral protein may be more easily discoverable than inhibitors, and though they may have weak or no antiviral activity, they can be used to develop PROTACs. In addition, as has been observed in the cancer biology field, targeted degradation may have key advantages over functional inhibition of drug targets, including improved selectivity[5,6] and potency[5,7,8], the ability to achieve efficacy against previously "undruggable" targets, as well as resilience to common resistance mechanisms, such as decreased drug-binding and target overexpression[9–12]. Due to limited application of TPD in antivirals development, however, the types of viral proteins that constitute good TPD targets remain poorly defined. Some viral proteins may not be good targets for TPD-mediated antiviral activity due to their subcellular localization and/or high expression level. In addition, the extent of depletion required to exert an antiviral effect likely varies depending on the viral protein targeted and its function(s) and mechanism(s) within the replication cycle of the virus.

Since prior efforts to develop PROTAC DAAs have largely focused on viral enzymes that execute their function of catalysis in a stoichiometric fashion, here we undertook development of PROTACs targeting a viral structural protein that functions as an oligomer. For this effort, we chose the DENV envelope (E) protein, which has essential functions during viral entry and in viral particle production that are genetically separable, providing the opportunity for E PROTACs to have two different antiviral modes of action. Conjugation of two known DENV E fusion inhibitors, GNF-2 and 2-12-2, to ligands of the substrate recognition subunit of the CRL4$^{CRBN}$ E3 ligase complex produced PROTACs GNF-2-deg and 2-12-2-deg, which cause proteasome-mediated degradation of E that is correlated with inhibition of DENV in

a cell culture model. Both E degraders exhibit antiviral potency significantly greater (4- to 10-fold) than that of the parental inhibitor (Fig. 3A), and this improvement is due to the TPD-based mechanism as it requires CRBN and is not observed under conditions in which the small molecule cannot interact with CRBN due to (i) modification of the CRBN ligand (GNF-2-deg-BUMP and 2-12-2-deg-BUMP in Figs. 2B, 3A), (ii) competition with excess CRBN ligand (Fig. 3B), or (iii) use of CRBN-deficient cells (Figs. 2B, 3A). While the generalizability of this phenomenon to other DENV E inhibitors or inhibitors of other viral fusion glycoproteins remains to be determined, our results empirically expand the repertoire of viral proteins demonstrated susceptible to TPD-mediated antiviral activity and also highlight several interesting aspects of targeting E.

## Small molecules targeting the E βOG pocket can exert antiviral activity through a dual mode of action

Prior efforts to develop DAAs targeting E have focused exclusively on inhibition of viral entry. Knowledge of E's structure and function prior to and during particle formation has been too limited to support rational design or screening for inhibitors of this process. In our earlier work developing GNF-2 and 2-12-2, we had also focused on E's function in entry, utilizing an infectivity assay in which compound treatment was restricted to a preincubation of the viral inoculum with compound and a one-hour infection window after which free compound was washed away and viral yield 24 h post-infection was used as the assay readout. This was to avoid the contribution of any intracellular off-targets (e.g., Abl kinases for GNF-2[40]) to our measurements of antiviral activity, but led us to overlook effects of GNF-2 and 2-12-2 on E's functions post-entry. Here we show that GNF-2 and 2-12-2 have effects on the abundance of intracellular E and secretion of viral particles in the VLP system, an experimental model that by-passes the viral entry, gene expression, and genome replication steps of the viral replication cycle. This post-entry antiviral activity is associated with notable increases in antiviral potency against some other mosquito-borne flaviviruses. For example, although GNF-2 exhibits concentration-dependent antiviral activity against DENV, ZIKV, JEV, and WNV Kunjin, the activity is modest when the compound-treatment is limited to the initial entry phase of the infectious cycle[31], but improves considerably when the compound is present throughout the post-entry

portion of single-cycle infection, with $EC_{90}$ values improving by 2- to over 10-fold against DENV, ZIKV, and JEV. GNF-2-deg and 2-12-2-deg also have a dual mode of action, inhibiting fusion during viral entry via a CRBN-independent mechanism that recapitulates the activity of the parental inhibitors (Supplementary Fig. 3) and reducing viral particle production via a CRBN-dependent mechanism (Fig. 4). While it is formally possible that the effect on viral particle production is due to a cellular target, this seems unlikely since it is observed with two different PROTACs derived from two structurally distinct E ligands. Decreased activity against the double E-F193L/M196V VLP mutant (Fig. 4D) also supports the idea that the effects of GNF-2-deg and 2-12-2-deg on viral particle production are due to binding at this site. Collectively, these results indicate that the βOG pocket of the flavivirus E protein is a molecular target that can mediate antiviral activity via two different modes of action and suggest that this dual mode of action can result in greater antiviral activity for both E inhibitors and E PROTACs.

## Mechanism of effects on intracellular E and particle production and implications for E's structure prior to budding

The effects of GNF-2-deg and 2-12-2-deg on intracellular E and DENV particle production are CRBN-, neddylation-, and proteasome-dependent, providing strong support for a TPD-mediated mechanism; however, this raises many questions regarding how ternary complex formation and proteasomal degradation of E are occurring. Since newly synthesized E has been thought to accumulate on the lumenal face of the ER membrane prior to virion budding[36,37,45], it is unclear how $CRL4^{CRBN}$, which is known to be cytosolic, accesses E as a substrate. Although the antiviral activity could formally be mediated by a cytosolic off-target that impacts viral particle formation, this seems unlikely because GNF-2-deg and 2-12-2-deg come from two structurally independent lead series; moreover, our proteomic profiling of the "degradeomes" of GNF-2-deg and 2-12-2-deg did not reveal a shared target known to affect flavivirus particle production[41]. Reduced activity against the E-F193L/M196V double-mutant also supports the idea that E is the relevant pharmacological target. Assuming that the effects of the two E PROTACs are "on-target," there are two possibilities to consider. First, small molecules may bind to E on the lumenal side of the ER membrane, with E subsequently relocalized (via an uncharacterized mechanism) to the cytosol where ternary complex formation and/or proteasomal degradation of E can occur. The observation of a pool of E that is not susceptible to degradation when GNF-2-deg and 2-12-2-deg are added at 16 h post-transfection in the VLP system or 12 h post-infection in the live virus system argues against this, as it suggests that E may no longer be susceptible to these PROTACs once it is on a viral particle that has entered the secretory system (Supplementary Fig. 6). A second hypothesis is that GNF-2-deg and 2-12-2-deg engage E before its insertion through the ER membrane, with ternary complex formation occurring while the protein is still on the cytosolic side of the ER membrane. In this case, the actual target may be the viral polyprotein, and small molecule-binding may potentially delay translocation across the membrane. The disappearance of other viral proteins (core, NS4B, NS5) in the presence of the E PROTACs (Supplementary Fig. 4) is consistent with this hypothesis although further experiments are needed to demonstrate polyprotein targeting explicitly and to rule out reduced viral gene expression due to general antiviral activity.

Related to this, the unanticipated effect of parental inhibitors GNF-2 and 2-12-2 on intracellular E and VLP secretion may also be consistent with targeting of E and/or the polyprotein on the cytosolic side of the ER membrane. In this case, GNF-2 and 2-12-2 may bind to newly synthesized E and induce its degradation by interfering with folding, dimerization, and/or translocation to the lumenal side of the ER membrane via a CRBN-independent mechanism. This may be reminiscent of the effects of selective estrogen receptor degraders (SERDs), which all bind to ERα but which induce its degradation through mechanisms that are distinguishable at the molecular level[46]. Additional work is needed to gain direct evidence for this and to interrogate the molecular details for the effects of both inhibitors and PROTACs on intracellular E. This includes foundational work characterizing the structure and topology of E as it exits the ribosome and as it accumulates on the lumenal side of the ER membrane prior to virion budding. Both the E inhibitors and E PROTACs may serve as useful probes for this effort.

## Targeted-protein degradation as an approach to address viral diversity

Collectively, flaviviruses represent a major genus of human pathogens that currently lack effective prevention and treatment strategies. The genetic and serologic diversity of flaviviruses has presented manifold challenges in both vaccine and antivirals development. Since the pharmacology of TPD-based drugs is event-driven rather than occupancy-driven, their potency is not strictly dictated by drug-target affinity or residence time. We have hypothesized that this potential to exert potent pharmacology despite modest affinities might enable TPD-based antivirals to tolerate point differences that limit the spectrum of activity of classical DAA inhibitors. Due to conservation of the βOG pocket, GNF-2 and 2-12-2, also have modest activity against ZIKV, WNV Kunjin, and JEV in viral entry assays[31] despite having been optimized against DENV. Conversion of these two E inhibitors to PROTACs resulted in CRBN-dependent effects on viral particle production as well as improved antiviral activity against the other flaviviruses tested with the exception of the activity of 2-12-2-deg against YFV. The effects of GNF-2-deg and 2-12-2-deg on intracellular E and VLP production are also unaffected by point mutations (E-F193, E-M196V, and E-F279S) that we previously showed reduce binding to $sE_2$ and susceptibility of DENV2 to GNF-2 and 2-12-2 in viral infectivity assays[31]. This work demonstrates that antiviral activity can be achieved by degrading non-enzymatic viral proteins, thus opening opportunities beyond targeting viral enzymes and highlights that E degraders have the potential to achieve broad-spectrum activity across multiple members of a viral genus. Along with our recently reported degrader molecule active against the hepatitis C virus NS3-4A protease and its drug resistant variants[13], our work supports implementation of TPD principles in antiviral drug discovery as a potential approach to broaden activity spectrum and/ or improve resistance profile.

## Methods

### Chemistry

**General methods.** Synthesis procedures followed standard protocols. All starting materials, reagents, and solvents were sourced from commercial suppliers and utilized without purification, unless otherwise specified. Reaction progress was monitored employing a Waters Acquity UPLC/MS system equipped with a Waters PDA eλ Detector, QDa Detector, Sample Manager - FL, and Binary Solvent Manager, coupled with an Acquity UPLC® BEH C18 column (2.1 × 50 mm, 1.7 µm particle size). The solvent gradient was set to 85% A at 0 min, shifting to 1% A at 1.7 min; where solvent A consisted of 0.1% formic acid in water and solvent B comprised 0.1% formic acid in acetonitrile, with a flow rate of 0.6 mL/min.

Purification of reaction products was conducted through flash column chromatography utilizing a CombiFlash®Rf system, employing Teledyne Isco RediSep® normal-phase silica flash columns (4 g, 12 g, 24 g, 40 g, or 80 g), and a Waters HPLC system equipped with a Sun-FireTM Prep C18 column (19 ×100 mm, 5 µm particle size). The solvent gradient was optimized to start with 80% A at 0 min, gradually transitioning to 10% A over 25 min, where solvent A comprised 0.035% TFA in water and solvent B contained 0.035% TFA in methanol, with a flow rate set to 20 mL/min.

Characterization of compounds was achieved through the acquisition of $^1$H NMR spectra employing 500 MHz Bruker Avance III spectrometers, and $^{13}$C NMR spectra recorded on 125 MHz Bruker Avance III spectrometers. Chemical shifts were referenced in parts per million (ppm, δ) relative to tetramethylsilane (TMS). Coupling constants (J) were reported in Hertz (Hz), while spin multiplicities were denoted as s (singlet), d (doublet), t (triplet), q (quartet), qd (quartet of doublets), dt (doublet of triplets), dd (doublet of doublets), ddd (doublet of doublets of doublets) or m (multiplet).

Combined extracts were washed with brine, dried with $Na_2SO_4$, then concentrated and purified by silica gel chromatography to obtain *tert*-butyl (2-(2-(2-((2-(2,6-dioxopiperidin-3-yl)-1,3-dioxoisoindolin-5-yl)oxy)ethoxy)ethoxy)ethyl)carbamate (125 mg, 45%). LC/MS (ESI) m/z calculated [M + H]$^+$ 506.21, found [M + H-Boc] 406.20.

*tert*-butyl (2-(2-(2-((2-(2,6-dioxopiperidin-3-yl)-1,3-dioxoisoindolin-5-yl)oxy)ethoxy)ethoxy)ethyl)carbamate (125 mg, 0.25 mmol) was

**Synthesis of 6-chloro-*N*-(4-(trifluoromethoxy)phenyl)pyrimidin-4-amine**

To the solution of 4,6-dichloro-pyrimidine (1.48 g, 10 mmol) and 4-trifluoromethoxyaniline (1.77 g, 10 mmol) in 50 mL of EtOH was added DIEA (1.9 mL, 11 mmol). The reaction mixture was stirred for 16 h at 80 °C. Then the mixture was concentrated and purified by silica gel column chromatography (30% EA/Hexane) to give the title compound **3** (2.54 g, 88% yield). LC/MS (ESI) m/z calculated [M + H]$^+$ 290.02, found 290.23/292.23.

**Synthesis of 3-(6-((4-(trifluoromethoxy)phenyl)amino)pyrimidin-4-yl)benzoic acid**

A mixture of 6-chloro-*N*-(4-(trifluoromethoxy)phenyl)pyrimidin-4-amine (1.5 g, 5.2 mmol), 3-carboxyphenylboronic acid (865 mg, 5.2 mmol), Pd(PPh$_3$)$_4$ (150 mg, 0.13 mmol), and 2 N sodium carbonate (7.8 mL, 15.6 mmol) in acetonitrile/water (v/v = 1/1, 30 mL) was heated to 80 °C under an nitrogen atmosphere. After refluxing for 6 h, the reaction mixture was filtered. The filtrate was cooled to room temperature and treated with 2 N HCl (pH is around 4) to afford 3-(6-(4-(trifluoromethoxy) phenylamino)pyrimidin-4-yl)benzoic acid as the yellow precipitate, which was collected and washed with water and air-dried to give 1.46 g of the title compound (75% yield). LC/MS (ESI) m/z calculated [M + H]$^+$ 376.08, found 376.34.

dissolved in 4 mL of DCM/TFA (1:1) and stirred for 2 h at room temperature before being concentrated and dried to provide 5-(2-(2-(2-aminoethoxy)ethoxy)ethoxy)-2-(2,6-dioxopiperidin-3-yl)isoindoline-1,3-dione. LC/MS (ESI) m/z calculated [M + H]$^+$ 406.15, found 406.20.

**Synthesis of *N*-(2-(2-(2-((2-(2,6-dioxopiperidin-3-yl)-1,3-dioxoisoindolin-5-yl)oxy)ethoxy)ethoxy)ethyl)-3-(6-((4-(trifluoromethoxy)phenyl)amino)pyrimidin-4-yl)benzamide (GNF-2-deg)**

5-(2-(2-(2-aminoethoxy)ethoxy)ethoxy)-2-(2,6-dioxopiperidin-3-yl)isoindoline-1,3-dione (100 mg, 0.25 mmol) and 3-(6-((4-(trifluoromethoxy)phenyl)amino)pyrimidin-4-yl)benzoic acid (93 mg, 0.25 mmol) were added to DMF (2.5 mL) followed by adding DIEA (216 μL, 1.2 mmol) and HATU (188 mg, 0.5 mmol) The reaction was stirred for 30 min and then purified by HPLC to provide the title compound (35 mg, 19%). LC/MS (ESI) m/z calculated [M + H]$^+$ 763.23, found 763.21.

$^1$H NMR (500 MHz, DMSO-$d_6$) δ 11.11 (s, 1H), 10.12 (s, 1H), 8.80 (d, J = 1.0 Hz, 1H), 8.71 (t, J = 5.6 Hz, 1H), 8.51 (t, J = 1.8 Hz, 1H), 8.16 (dt, J = 7.8, 1.4 Hz, 1H), 8.01 (dt, J = 7.8, 1.4 Hz, 1H), 7.87 − 7.83 (m, 2H), 7.80 (d, J = 8.3 Hz, 1H), 7.64 (t, J = 7.8 Hz, 1H), 7.41 (d, J = 2.3 Hz, 1H),

**Synthesis of 5-(2-(2-(2-aminoethoxy)ethoxy)ethoxy)-2-(2,6-dioxopiperidin-3-yl)isoindoline-1,3-dione**

2-(2,6-dioxopiperidin-3-yl)-5-hydroxyisoindoline-1,3-dione (150 mg, 0.55 mmol) was dissolved in DMF (3 mL), treated with K$_2$CO$_3$ (151 mg, 1.1 mmol), *tert*-butyl (2-(2-(2-bromoethoxy)ethoxy)ethyl)carbamate (171 mg, 0.55 mmol) was added, and the mixture stirred 6 h at 50 °C. The reaction was diluted with water and extracted with EtOAc.

7.40−7.36 (m, 2H), 7.34 − 7.31 (m, 2H), 5.11 (dd, J = 12.8, 5.4 Hz, 1H), 4.29 − 4.25 (m, 2H), 3.82 − 3.76 (m, 2H), 3.66 − 3.56 (m, 6H), 3.47 (q, J = 5.8 Hz, 2H), 2.89 (ddd, J = 16.8, 13.7, 5.4 Hz, 1H), 2.63 − 2.52 (m, 2H), 2.08−2.01 (m, 1H).

## Synthesis of 5-(2-(2-(2-aminoethoxy)ethoxy)ethoxy)-2-(2-oxopiperidin-3-yl)isoindoline-1,3-dione

A mixture of 5-hydroxyisobenzofuran-1,3-dione (1 g, 6.1 mmol), 3-aminopiperidin-2-one hydrochloride (1 g, 6.7 mmol), and NaOAc (1.8 g, 18.6 mmol) was dissolved in acetic acid (20 mL), and the resulting mixture was stirred at 90 °C for 16 h. After the reaction was completed, the solution was filtered and the filtration residue was washed by water and dried to yield a yellow solid 5-hydroxy-2-(2-oxopiperidin-3-yl)iso-

[1]H NMR (500 MHz, DMSO-$d_6$) δ 9.92 (s, 1H), 8.76 (d, J = 1.1 Hz, 1H), 8.70 (t, J = 5.6 Hz, 1H), 8.52 (t, J = 1.8 Hz, 1H), 8.18 (dt, J = 7.8, 1.5 Hz, 1H), 7.99 (dt, J = 7.8, 1.4 Hz, 1H), 7.86 (d, J = 2.2 Hz, 1H), 7.84 (d, J = 3.2 Hz, 2H), 7.76 (s, 1H), 7.63 (t, J = 7.8 Hz, 1H), 7.37 (d, J = 5.5 Hz, 1H), 7.36 (d, J = 1.7 Hz, 2H), 7.32 (d, J = 1.2 Hz, 1H), 7.29 (dd, J = 8.3, 2.4 Hz, 1H), 4.56 (dd, J = 12.0, 6.3 Hz, 1H), 4.27 – 4.24 (m, 2H), 3.81 – 3.76 (m, 2H), 3.66 – 3.56 (m, 6H), 3.47 (q, J = 5.9 Hz, 2H), 3.28 – 3.15 (m, 2H), 2.21 (qd, J = 12.2, 4.8 Hz, 1H), 2.02 – 1.94 (m, 1H), 1.93 – 1.78 (m, 2H).

indoline-1,3-dione, which was used in the next step without further purification (870 mg, 55%).

5-hydroxy-2-(2-oxopiperidin-3-yl)isoindoline-1,3-dione (50 mg, 0.17 mmol) was dissolved in DMF (1.5 mL), treated with K$_2$CO$_3$ (46 mg, 0.34 mmol), tert-butyl (2-(2-(2-bromoethoxy)ethoxy)ethyl)carbamate (55 mg, 0.17 mmol) was added, and the mixture stirred 6 h at 50 °C. The reaction was diluted with water and extracted with EtOAc. Combined extracts were washed with brine, dried with Na$_2$SO$_4$, then concentrated and purified by silica gel chromatography to obtain tert-butyl (2-(2-(2-((1,3-dioxo-2-(2-oxopiperidin-3-yl)isoindolin-5-yl)oxy)ethoxy)ethoxy)ethyl)carbamate (31 mg, 38%). LC/MS (ESI) m/z calculated [M + H]$^+$ 492.23, found [M + H-Boc] 392.45

tert-butyl (2-(2-(2-((1,3-dioxo-2-(2-oxopiperidin-3-yl)isoindolin-5-yl)oxy)ethoxy)ethoxy)ethyl) carbamate (31 mg, 0.063 mmol) was dissolved in 2 mL of DCM/TFA (1:1) and stirred for 2 h at room temperature before being concentrated and dried to provide 5-(2-(2-(2-aminoethoxy)ethoxy)ethoxy)-2-(2-oxopiperidin-3-yl)isoindoline-1,3-dione without further purification. LC/MS (ESI) m/z calculated [M + H]$^+$ 392.17, found 392.45.

## Synthesis of 2-chloro-N-(4-(trifluoromethoxy)phenyl)pyrimidin-4-amine

To the solution of 2,4-dichloropyrimidine (1.48 g, 10 mmol) and 4-trifluoromethoxyaniline (1.77 g, 10 mmol) in 50 mL of EtOH was added DIEA (1.9 mL, 11 mmol). The reaction mixture was stirred for 16 h at 80 °C. Then the mixture was concentrated and purified by silica gel column chromatography (30% EA/Hexane) to give 2-chloro-N-(4-(trifluoromethoxy)phenyl)pyrimidin-4-amine (2.4 g, 83% yield). LC/MS (ESI) m/z calculated [M + H]$^+$ 290.02, found 290.24/292.19

## Synthesis of 4-((4-((4-(trifluoromethoxy)phenyl)amino)pyrimidin-2-yl)amino)-3-(trifluoromethyl)benzoic acid

A mixture of 2-chloro-N-(4-(trifluoromethoxy)phenyl)pyrimidin-4-amine (550 mg, 19 mmol), 4-amino-3-(trifluoromethoxy)benzoic acid (420 mg, 1.9 mmol), and TFA (190 μL, 3.8 mmol) in s-butanol (6 mL) was heated to 100 °C. After refluxing for 6 h, the reaction was diluted with water and extracted with EtOAc. Combined extracts were washed

**ZXH-2-107-Neg
(GNF-2-deg-BUMP)**

## Synthesis of N-(2-(2-(2-((1,3-dioxo-2-(2-oxopiperidin-3-yl)isoindolin-5-yl)oxy)ethoxy)ethoxy)ethyl)-3-(6-((4-(trifluoromethyl)phenyl)amino)pyrimidin-4-yl)benzamide (GNF-2-deg-BUMP)

with brine, dried with Na$_2$SO$_4$, then concentrated and purified by silica gel chromatography to obtain 4-((4-((4-(trifluoromethoxy)phenyl)amino)pyrimidin-2-yl)amino)-3-(trifluoromethyl)benzoic acid (380 mg, 42%).LC/MS (ESI) m/z calculated [M + H]$^+$ 475.08, found 475.42.

**ZXH-8-004
(2-12-2-deg)**

5-(2-(2-(2-aminoethoxy)ethoxy)ethoxy)-2-(2-oxopiperidin-3-yl)isoindoline-1,3-dione (30 mg, 0.076 mmol) and 3-(6-(4-(trifluoromethoxy) phenylamino)pyrimidin-4-yl)benzoic acid (29 mg, 0.064 mmol) were added to DMF (2.0 mL) followed by addition of DIEA (66 μL, 0.38 mmol) and HATU (60 mg, 0.152 mmol) The reaction was stirred for 30 min and then purified by HPLC to provide the title compound (26 mg, 32%). LC/MS (ESI) m/z calculated [M + H]$^+$ 749.25, found 749.78.

## Synthesis of 4-(2-(2-(2-aminoethoxy)ethoxy)ethoxy)-2-(2,6-dioxopiperidin-3-yl)isoindoline-1,3-dione

2-(2,6-dioxopiperidin-3-yl)-4-hydroxyisoindoline-1,3-dione (100 mg, 0.36 mmol) was dissolved in DMF (3 mL), treated with K$_2$CO$_3$ (101 mg, 0.73 mmol), tert-butyl (2-(2-(2-bromoethoxy)ethoxy)ethyl)carbamate (125 mg, 0.41 mmol) was added, and the mixture stirred 6 h at 50 °C. The reaction was diluted with water and extracted with EtOAc.

Combined extracts were washed with brine, dried with $Na_2SO_4$, then concentrated and purified by silica gel chromatography to obtain *tert*-butyl (2-(2-(2-((2-(2,6-dioxopiperidin-3-yl)-1,3-dioxoisoindolin-4-yl) oxy)ethoxy)ethoxy)ethyl)carbamate (42 mg, 23%). LC/MS (ESI) m/z calculated $[M + H]^+$ 506.21, found $[M + H\text{-Boc}]$ 406.20.

*tert*-butyl (2-(2-(2-((2-(2,6-dioxopiperidin-3-yl)-1,3-dioxoisoindolin-4-yl)oxy)ethoxy)ethoxy)ethyl)carbamate (42 mg, 0.08 mmol) was dissolved in 3 mL of DCM/TFA (1:1) and stirred for 2 h at room temperature before being concentrated and dried to provide 5-(2-(2-(2-aminoethoxy)ethoxy)ethoxy)-2-(2,6-dioxopiperidin-3-yl)isoindoline-1,3-dione without further purification. LC/MS (ESI) m/z calculated $[M + H]^+$ 406.15, found 406.20.

### Synthesis of *N*-(2-(2-(2-((2-(2,6-dioxopiperidin-3-yl)-1,3-dioxoisoindolin-4-yl)oxy)ethoxy)ethoxy)ethyl)-3-(trifluoromethoxy)-

**19** → **ZXH-8-004-Neg (2-12-2-deg-BUMP)**

14, HATU, DIEA
DMF, r.t.

### 4-((4-((4-(trifluoromethoxy)phenyl)amino)pyrimidin-2-yl) amino)benzamide (2-12-2-deg)

4-(2-(2-(2-aminoethoxy)ethoxy)ethoxy)-2-(2,6-dioxopiperidin-3-yl)iso-indoline-1,3-dione (27 mg, 0.067 mmol) and 4-((4-((4-(tri-fluoromethoxy)phenyl)amino)pyrimidin-2-yl)amino)-3-(tri-fluoromethyl)benzoic acid (32 mg, 0.067 mmol) were added to DMF (1 mL) followed by addition of DIEA (60 μL, 0.34 mmol) and HATU (51 mg, 0.13 mmol) The reaction stirred for 30 min and then purified by HPLC to provide the title compound (30 mg, 52%). LC/MS (ESI) m/z calculated $[M + H]^+$ 862.22, found 862.38. $^1$H NMR (500 MHz, DMSO-$d_6$) δ 11.10 (s, 1H), 10.08 (s, 1H), 9.31 (s, 1H), 8.68 (t, J = 5.6 Hz, 1H), 8.10 – 8.05 (m, 2H), 7.92 – 7.86 (m, 2H), 7.79 (dd, J = 8.5, 7.3 Hz, 1H), 7.74 – 7.67 (m, 2H), 7.50 (d, J = 8.5 Hz, 1H), 7.45 (d, J = 7.2 Hz, 1H), 7.27 (d, J = 8.1 Hz, 1H), 6.39 (d, J = 6.2 Hz, 1H), 5.08 (dd, J = 12.8, 5.4 Hz, 1H), 4.35 – 4.29 (m, 1H), 3.83 – 3.78 (m, 2H), 3.67 (dd, J = 5.9, 3.6 Hz, 2H), 3.61 – 3.53 (m, 4H), 3.44 (q, J = 5.8 Hz, 2H), 2.88 (ddd, J = 16.8, 13.8, 5.4 Hz, 1H), 2.63 – 2.52 (m, 2H), 2.50 – 2.47 (m, 2H), 2.06 – 1.99 (m, 1H).

4-hydroxy-2-(2-oxopiperidin-3-yl)isoindoline-1,3-dione (50 mg, 0.17 mmol) was dissolved in DMF (1.5 mL), treated with $K_2CO_3$ (46 mg, 0.34 mmol), *tert*-butyl (2-(2-(2-bromoethoxy)ethoxy)ethyl)carbamate (55 mg, 0.17 mmol) was added, and the mixture stirred 6 h at 50 °C. The reaction was diluted with water and extracted with EtOAc. Combined extracts were washed with brine, dried with $Na_2SO_4$, then concentrated and purified by silica gel chromatography to obtain *tert*-butyl (2-(2-(2-((1,3-dioxo-2-(2-oxopiperidin-3-yl)isoindolin-4-yl)oxy)ethoxy)ethoxy) ethyl)carbamate (29 mg, 35%). LC/MS (ESI) m/z calculated $[M + H]^+$ 492.23, found $[M + H\text{-Boc}]$ 392.45.

*tert*-butyl (2-(2-(2-((1,3-dioxo-2-(2-oxopiperidin-3-yl)isoindolin-4-yl)oxy)ethoxy)ethoxy)ethyl) carbamate (29 mg, 0.06 mmol) was dissolved in 2 mL of DCM/TFA (1:1) and stirred for 2 h at room temperature before being concentrated and dried to provide 4-(2-(2-(2-aminoethoxy)ethoxy)ethoxy)-2-(2-oxopiperidin-3-yl)isoindoline-1,3-dione. LC/MS (ESI) m/z calculated $[M + H]^+$ 392.17, found 392.45.

### Synthesis of *N*-(2-(2-(2-((1,3-dioxo-2-(2-oxopiperidin-3-yl)iso-indolin-4-yl)oxy)ethoxy)ethoxy)ethyl)-3-(trifluoromethoxy)-4-((4-((4-(trifluoromethoxy)phenyl)amino)pyrimidin-2-yl)amino) benzamide (2-12-2-deg-BUMP)

4-(2-(2-(2-aminoethoxy)ethoxy)ethoxy)-2-(2-oxopiperidin-3-yl)iso-indoline-1,3-dione (25 mg, 0.064 mmol) and 4-((4-((4-(tri-fluoromethoxy)phenyl)amino)pyrimidin-2-yl)amino)-3-(tri-fluoromethyl)benzoic acid (32 mg, 0.064 mmol) (29 mg, 0.064 mmol) were added to DMF (1 mL) followed by addition of DIEA (55 μL, 0.32 mmol) and HATU (50 mg, 0.128 mmol) The reaction stirred for 30 min and then purified by HPLC to provide the title compound (26 mg, 48%). LC/MS (ESI) m/z calculated $[M + H]^+$ 848.24, found 848.93. $^1$H NMR (500 MHz, DMSO-$d_6$) δ 10.60 (s, 1H), 9.96 (s, 1H), 8.74 (t, J = 5.6 Hz, 1H), 8.08 (d, J = 6.7 Hz, 1H), 7.99 (d, J = 8.4 Hz, 1H), 7.93 (dd, J = 11.7, 3.2 Hz, 2H), 7.83 (d, J = 3.0 Hz, 1H), 7.76 (dd, J = 8.5, 7.3 Hz, 1H), 7.64 (d, J = 8.7 Hz, 2H), 7.47 (d, J = 8.5 Hz, 1H), 7.41 (d, J = 7.2 Hz, 1H), 7.28

**17** + **9** → **18** → **19**

KOAc, AcOH, 90 °C

1.$K_2CO_3$, DMF, BrPEG₂NHBoc, 50 °C
2. TFA, DCM, r.t.

### Synthesis of 4-(2-(2-(2-aminoethoxy)ethoxy)ethoxy)-2-(2-oxopi-peridin-3-yl)isoindoline-1,3-dione

A mixture of 4-hydroxyisobenzofuran-1,3-dione (1 g, 6.1 mmol), 3-aminopiperidin-2-one hydrochloride (1 g, 6.7 mmol), and NaOAc (1.8 g, 18.6 mmol) was dissolved in HOAc (20 mL), and the resulting mixture was stirred at 90 °C for 16 h. After the reaction was completed, the solution was filtered and the filtration residue was washed by water and dried to yield a yellow solid, which was used in the next step without further purification (950 mg, 60%).

(d, J = 8.6 Hz, 2H), 6.46 (d, J = 6.7 Hz, 1H), 4.54 (dd, J = 12.0, 6.3 Hz, 1H), 4.30 (t, J = 4.5 Hz, 2H), 3.82 – 3.77 (m, 2H), 3.67 (dd, J = 5.9, 3.6 Hz, 2H), 3.61 – 3.54 (m, 4H), 3.45 (q, J = 5.8 Hz, 2H), 3.21 (dt, J = 10.1, 5.0 Hz, 2H), 2.20 (qd, J = 12.2, 4.2 Hz, 1H), 2.03 – 1.93 (m, 1H), 1.91 – 1.84 (m, 2H).

### Cell culture

The mammalian cell lines used in the study were maintained in Dul-becco's Modified Eagle's medium (DMEM) supplemented with non-essential amino acids, 10 mM Hepes, and 10% fetal bovine serum (FBS)

at 37 °C with 5% CO$_2$. Huh7.5 cells are obtained from Charles Rice (Rockefeller University). Baby Hamster Kidney cells (BHK) are obtained from Eva Harris (University of California, Berkeley). Vero cells are purchased from ATCC. C6/36 cells, a mosquito cell line derived from Aedes albopictus (Diptera: Culicidae) embryonic tissue are purchased from ATCC and cultured in Leibovitz medium (L-15) containing 10% FBS at 28 °C.

## Viruses

Dengue virus serotype 2 New Guinea C (DENV2 NGC) was obtained from Lee Gehrke (Massachusetts Institute of Technology). Zika virus (ZIKA) strain PF-25013-18 was obtained from Didier Musso (Institut Louis Malardé, French Polynesia). Japanese encephalitis virus (JEV) vaccine strain SA14-14-2 was obtained from Gregory Gromowski (Walter Reed Army Institute of Research). West Nile virus strain Kunjin (WNV) was obtained from Michael Diamond (Washington University). These viruses are propagated in C6/36 cells as previously described[31,40]. Yellow fever virus strain Kouma was obtained from the World Reference Center for Emerging Viruses and Arboviruses (University of Texas Medical Branch). The YFV stock was prepared by passaging once in Vero cells cultured in DMEM with 2% FBS. The supernatant was collected 3 days post-inoculation upon observation of cytopathic effects. Cell debris was removed by centrifugation and filtration. Virus stocks were aliquoted and stored at −80 °C. All work with DENV2 NGC, ZIKA, JEV, and WNV was performed in a biosafety level 2 laboratory (BSL2) laboratory with BSL2 enhanced procedures. Experiments involving YFV are performed in a biosafety level 3 laboratory (BSL3) at the University of Wisconsin-Madison by vaccinated personnel equipped with a powered air purifying respirator and appropriate personal protective equipment. All work was reviewed and approved by the institutional biosafety committee at the institution where the work was conducted (Stanford Administrative Panel on Biosafety and Harvard Committee on Microbiological Safety for DENV, WNV Kunjin, ZIKV, JEV) and University of Wisconsin-Madison Biosafety Committee (YFV).

## Immunoblotting and antibodies

Monoclonal antibody 4G2 targeting E protein was collected from culture medium of hybridoma D1-4G2-4-15 (ATCC HB-112) and used at a ratio of 1:100 for immunoblotting of DENV2 NCG, ZIKA, WNV E. Rabbit polyclonal antibody anti-JEV E protein was purchased from GeneTex (GTX125867) and used at a ratio of 1:1000 for immunoblotting of JEV E. Mouse monoclonal antibody anti-GAPDH was purchased from Gene-Tex (GTX28245) and used at a ratio of 1:10000. Horseradish peroxidase (HRP)-conjugated goat anti-mouse IgG was purchased from Bio-Rad Laboratories (170-6516) and used at a ratio of 1:3000. HRP-conjugated goat anti-rabbit IgG antibodies were purchased from Bio-Rad Laboratories (170-6515) and used at a ratio of 1:3000.

## Antiviral assay

Huh7.5 cells were seeded into 24-well plates at 50,000 cells per well and incubated for 24 h. Cells were infected with virus (DENV2 NGC, ZIKA, JEV, or WNV) at a MOI of 1 for 1 h and washed with PBS to remove the extracellular virus. The infected cells were then treated with compounds at varying concentrations in DMEM supplemented with nonessential amino acids and 2% FBS. At 24 h post-infection, the cell lysates were collected for characterization of protein abundance via immunoblotting, and the culture supernatants were collected for quantification of viral titer via plaque formation assay as previously described[31,32].

To assess antiviral activity against YFV, Huh7.5 cells were seeded into 96-well plates at 40,000 per well and incubated for 24 h. The cells were then infected with YFV at an MOI of 1 for 1 h, after which they were washed with Dulbecco's Phosphate Buffered Saline (DPBS) to remove the extracellular virus. The infected cells were then treated with compounds at varying concentrations in DMEM supplemented with 2% FBS. At 24 h post-infection, the culture supernatants were collected and subjected to quantification of viral titer via focus-forming assay. Briefly, 50,000 Vero cells are seeded into 96 well plates and incubated for 24 h. The cells were then incubated with 100 µL of a tenfold serial dilution of culture supernatants for 1 h at 37 °C and then washed with DPBS. 100 µL of DMEM supplemented with 2% FBS and subsequently, 125 µL of methylcellulose overlay were added into each well. Following incubation for 48 h at 37 °C, cells were fixed with 4% paraformaldehyde in DPBS for 30 min at room temperature, washed with DPBS, and then incubated overnight with monoclonal antibody 2D12 mouse anti-YFV E (CRL-1689, ATCC) at a ratio of 1:800 at 4 °C and washed with DPBS. After washing, the cells were incubated with HRP-conjugated goat anti-mouse antibody (115-035-062, Jackson ImmunoResearch) at a ratio of 1:1000 for 1 h at room temperature. The foci were then visualized by incubation with TrueBlue™ peroxidase substrate (#50-78-02, KPL) and quantified by BioSpot plate reader (ImmunoSpot, Cellular Technology).

## Virus-like particle (VLP) assay

The plasmid used in the study was obtained from Stephen C. Harrison (Harvard Medical School) and has a codon-optimized sequence of DENV2-FGA/02 prM-E incorporated into a pcDNA3.1 backbone[43]. The plasmid was mutagenized using the primers listed in Supplementary Table 1 and following the instruction of Quikchange II site-directed mutagenesis kits (Agilent).

To assay antiviral activity in the VLP system, 100,000 Huh7.5 cells were seeded into each well of a 12-well plate for 24 h prior to transfection (2 µg of plasmid with 4 µL of lipofectamine 2000 (Life Technologies)). At 4 h post-transfection, the medium was replaced with DMEM supplemented with 2% FBS and the indicated concentration of compounds. Following 24 h compound treatment, the supernatant and the cell lysates were collected. The VLPs were isolated from 500 µL supernatant by loading over a 500 µL sucrose cushion (12.5% sucrose, 10 mM Tris-HCl, 2.5 mM EDTA, 50 mM NaCl at pH7.4) followed by centrifugation at $100,000 \times g$ for 2.5 h at 4 °C. Immunoblotting for E was performed to quantify intracellular E in lysates and VLPs in the isolated preparations.

## Reporting summary

Further information on research design is available in the Nature Portfolio Reporting Summary linked to this article.

## Data availability

Datasets generated and/or analyzed during the current study are included in the paper or are appended as supplementary data. The source data underlying Figs. 2B, D–F, 3A–D, 4A, D, 5 and Supplementary Figs. 1, 2, 3B, 4A–B, 5, 6B, D are provided as a Source Data file. Source data are provided with this paper.

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

## Acknowledgements

This work was supported by NIH awards R01AI48632 and R01AI146152 and a Mark Foundation Emerging Leader Award 19-001-ELA (grant to E.S.F). We thank the following colleagues for sharing reagents: Gregory

Gromowski for JEV SA14-14-2; Didier Musso and Nathalie Pardigon for ZIKV PF13-251013-18; Lee Gehrke for DENV2 New Guinea C virus; Eva Harris for BHK-21 cells; Charles Rice for Huh7.5 cells; Stephen C. Harrison for the DENV VLP plasmid. We thank Milka Kostic and members of the Yang and Gray laboratories for helpful discussions. Schematics in Figs. 2A, C, and Supplementary Figs. 4A, B were created in BioRender and are published here under a CC-BY-NC-ND license. Open access to this article does not include the use of the images created in BioRender.

## Author contributions

H.L., Z.L., N.S.G., and P.L.Y. conceived the project. H.L., Z.L, T.R., X.Q., K.A.D, A.B., E.S.F., T.Z., N.S.G., and P.L.Y. designed experiments. Z.L., Z.H., and R.P.G. performed chemical syntheses. H.L., T.R., X.Q., and K.A.D, performed biological evaluation experiments and analyzed data. H.L., Z.L., N.S.G., and P.L.Y. wrote the paper with input from all authors. All authors reviewed and edited the paper. H.L. and Z.L. contributed equally to the work.

## Competing interests

N.S.G is a founder, science advisory board member (SAB) and equity holder in Syros, C4, Allorion, Lighthorse, Voronoi, Inception, Matchpoint, CobroVentures, GSK, Shenandoah (board member), Larkspur (board member) and Soltego (board member). The Gray lab receives or has received research funding from Novartis, Takeda, Astellas, Taiho, Jansen, Kinogen, Arbella, Deerfield, Springworks, Interline and Sanofi. E.S.F. is a founder, member of the scientific advisory board (SAB), and equity holder of Civetta Therapeutics, Lighthorse, Proximity Therapeutics, and Neomorph Inc (also board of directors), SAB member and equity holder in Avilar Therapeutics and Photys Therapeutics, and a consultant to Astellas, Sanofi, Novartis, Deerfield, Ajax and EcoR1 capital. The Fischer laboratory receives or has received research funding from Novartis, Deerfield, Ajax, Interline, and Astellas. K.A.D receives or has received consulting fees from Kronos Bio and Neomorph Inc. T.Z. is a scientific founder, equity holder, and consultant of Matchpoint, equity holder of Shenandoah. H.L. is currently an employee of Amgen. The remaining authors declare no competing interests.
