## [Peer Review File · Nature Communications]

REVIEWER COMMENTS

Reviewer #1 (Remarks to the Author):

This is an interesting study describing the development of PROTACs targeting the dengue virus envelope protein by linking E fusion inhibitors to CRBN recruiters. The resulting PROTACs block viral entry through inhibition of E-mediated membrane fusion and block viral production. Interestingly, the reported PROTACs also exhibited promising activity across a panel of other mosquito-borne flaviviruses. The study is very rigorously performed and the data are convincing. One minor comment is that the figure annotations are very difficult to read because of the small font sizes in Figure 2, 3, and 5. Could these be enlarged to enable better reading?

Reviewer #2 (Remarks to the Author):

The manuscript by Liu and colleagues probes the activity and mechanism of action of a new type of targeted antiviral against dengue virus. The authors explore the use of proteolysis targeting chimeras (PROTACs) directed at the dengue virus E protein. The molecule is built on previous work of small molecules that bind to a region of E protein and inhibit viral fusion. This is used as a targeting domain which is then coupled to a ligand of cereblon resulting in targeting by E3 ubiquitin ligase and subsequent degradation. Thus, there are two antiviral activities with one blocking viral entry and the second interfering with virion production through depletion of E protein. The authors demonstrate that the E depletion activity is dependent on CRBN-mediated degradation pathways. This ability to deplete E, which is shown to be dose-dependent, is an effective strategy to inhibit virus infection. This work is an important advance and the experiments are well presented and executed. Clearly this may be an important new target for virus intervention. There are several points that the authors should address.

Specific points:

Figure 2: The authors clearly show the depletion of dengue E protein and its dependence on CRBN and the proteasome in the presence of their PROTACs. The authors should demonstrate that other viral proteins such as capsid and/or NS5 are not depleted with this treatment. This would further demonstrate that RNA synthesis and virus protein translation are not impacted. Additionally, RNA synthesis could be measured to demonstrate that inhibition is at the level of E protein removal, and not affecting other steps in virus replication.

Two minor points on Figure 2: Why was an MOI of one chosen since not every cell will be simultaneously infected? Why do the authors graph the X-axis with a decreasing concentration of PROTAC rather than increasing?

Figure 3: It is interesting that the authors chose to measure virus titer as % DMSO. I find this difficult to follow. What is the absolute reduction in infectious viral units of the PROTACs inhibition?

Figure 4: It would be useful if the bands seen in Panels A and D were quantified for easier comparison.

Line 441: Have the authors considered to assess whether particles are budding into the ER lumen and thus degradation of E comes at a later step?

REVIEWER COMMENTS

Reviewer #1 (Remarks to the Author):

This is an interesting study describing the development of PROTACs targeting the dengue virus envelope protein by linking E fusion inhibitors to CRBN recruiters. The resulting PROTACs block viral entry through inhibition of E-mediated membrane fusion and block viral production. Interestingly, the reported PROTACs also exhibited promising activity across a panel of other mosquito-borne flaviviruses. The study is very rigorously performed and the data are convincing. One minor comment is that the figure annotations are very difficult to read because of the small font sizes in Figure 2, 3, and 5. Could these be enlarged to enable better reading?

Response: We thank the reviewer for the positive assessment of this work. We have revised our figures to enlarge the font size as recommended.

Reviewer #2 (Remarks to the Author):

The manuscript by Liu and colleagues probes the activity and mechanism of action of a new type of targeted antiviral against dengue virus. The authors explore the use of proteolysis targeting chimeras (PROTACs) directed at the dengue virus E protein. The molecule is built on previous work of small molecules that bind to a region of E protein and inhibit viral fusion. This is used as a targeting domain which is then coupled to a ligand of cereblon resulting in targeting by E3 ubiquitin ligase and subsequent degradation. Thus, there are two antiviral activities with one blocking viral entry and the second interfering with virion production through depletion of E protein. The authors demonstrate that the E depletion activity is dependent on CRBN-mediated degradation pathways. This ability to deplete E, which is shown to be dose-dependent, is an effective strategy to inhibit virus infection. This work is an important advance and the experiments are well presented and executed. Clearly this may be an important new target for virus intervention. There are several points that the authors should address.

Response: We appreciate the reviewer's positive evaluation of the importance of our work and for the comments and suggestions, which have helped us to improve the manuscript. We have responded to each of the reviewer's suggestions/comments below.

Specific points:

(1) Figure 2: The authors clearly show the depletion of dengue E protein and its dependence on CRBN and the proteasome in the presence of their PROTACs. The authors should demonstrate that other viral proteins such as capsid and/or NS5 are not depleted with this treatment. This would further demonstrate that RNA synthesis and virus protein translation are not impacted. Additionally, RNA synthesis could be measured to demonstrate that inhibition is at the level of E protein removal, and not affecting other steps in virus replication.

Response: The reviewer's request prompted us to think about this and to conduct additional experiments. In brief, when we repeated the analysis to monitor abundance of

E, core, NS4B, and NS5 at 24 hours post-infection, we observed depletion of all four proteins by GNF-2-deg and 2-12-2-deg. We can think of at least two non-mutually exclusive explanations for this that are both consistent with the E-specific and CRBN-dependent activity of GNF-2-deg and 2-12-2-deg in the VLP and live virus assays. First is that the 24 hour post-infection (hpi) time point at which we made these measurements is late enough for reinfection to occur. Reduced viral yield from the initial cycle of infection reduces the rate of reinfection, with add-on effects on viral translation and genome replication due to decreased template that affect the abundance of all of the viral proteins. In addition, the inhibitory effect of both the E inhibitors and the E degraders on fusion during this second cycle of infection would be expected to have an effect on all of the viral proteins as time extends beyond the initial cycle of infection. Finally, any antiviral activity exerted through an off-target effect would also eventually lead to a decrease in all of the viral proteins. We will refer to these potential causes as “general antiviral activity” since it is not caused directly by targeted protein degradation induced by the E degraders. The second explanation we considered is that the E degraders may be targeting E within the context of the polyprotein, which could lead to degradation of core, NS4B, and NS5.

Since our original experiments examined compound treatment over the 1-24 hpi timeframe, we analyzed earlier time points. Our reasoning was that if GNF2-deg and 2-12-2-deg target E that has already been cleaved from the polyprotein, then we should be able to observe its depletion earlier than we can detect depletion of the other viral proteins and before depletion of viral proteins due to general antiviral activity affecting a second round of infection impacts our observations. For these experiments, we chose a single concentration of 2.5 μM , which approximates the EC_{90} values of GNF-2-deg and 2-12-2-deg (3.5 μM and 1.7 μM , respectively) measured at MOI of 1 and 1-24 hpi treatment (data in Figure 3 in the manuscript). We observed depletion of core, NS4B, and NS5 alongside E at 20 and 22 hpi. Representative data from these experiments are shown in **Response Figure 1A** below. We were unable to obtain meaningful data at time points earlier than this because we could not consistently detect the viral proteins by Western blot at 18 hpi or earlier, even when we increased the MOI to 5 (data not shown). While it is possible that a second round of infection is occurring at 20 hpi and contributing to the template pool for viral translation, the magnitude of this effect seems unlikely to account for the significant reductions in core, NS4B, and NS5 that we observe. We also think that if reinfection and general antiviral activity were major reasons for the reductions in core, NS4B, and NS5, then we would also observe these proteins depleted in the presence of the parental inhibitors, GNF-2 and 2-12-2, as well as in the presence of the GNF-2-deg-BUMP and 2-12-2-deg-BUMP negative controls.

With respect to the potential of the E degraders to act on the polyprotein, since the original experiments evaluated compound treatment for hours 1-24 post-infection, we also conducted experiments in which we delayed treatment with the compounds to examine whether the E degraders are targeting E early in the replication cycle (*i.e.*, prior to budding and possibly in the viral polyprotein) versus late in the replication cycle after budding has occurred (**Figure 1B**, below). Again utilizing 2.5 μM GNF-2-deg and 2-12-2-deg, we found that when compound treatment is initiated at 5hpi, E is depleted along with core, NS4B, and NS5 proteins. When compound treatment was delayed to 11 hpi, we no longer observed depletion of E. Working with the VLP system, we also found the degraders'

effect on intracellular E was greatest when GNF2-deg and 2-12-2-deg were added earlier than 10 hours post-transfection (**Response Figure 2**). Incomplete degradation of a target protein by a PROTAC often reflects the existence of a subcellular pool of the target that is no longer accessible to the PROTAC and/or the UPS machinery. We believe that the E detected in cell lysates when compound treatment is delayed until 11-24hpi (or post-transfection in the VLP experiments) reflects E already in particles that have budded into the secretory system and that are no longer accessible to the UPS machinery required for the TPD mechanism.

Response Figure 1. GNF-2-deg and 2-12-2-deg cause depletion of other viral proteins in DENV in infected cells. (A) Schematic of experiment. Cells were infected with DENV2 at an MOI of 1 and compound treatment (2.5 μ M) initiated at 1hpi. Cell lysates were harvested at 14 to 24-hpi for analysis of E, core NS4B, and NS5 by Western blot. Western blots show time-dependent depletion of E, NS5, NS4B, and core in the presence of GNF-2-deg and 2-12-2-deg at 20, 22, and 24hpi. Depletion of viral proteins is not observed in the presence of the parental inhibitors, GNF-2 and 2-12-2, or in the presence of negative control compounds GNF-2-deg-BUMP or 2-12-2-deg-BUMP. Timepoints earlier than 20h did not result in consistent detection of viral proteins (data not shown). The representative results are shown from $n = 2$ independent experiments. (B) Schematic of experiment. Cells were infected with DENV2 at an MOI of 1, and compound-treatment (2.5 μ M) initiated at 1 to 11 hpi. Cell lysates were harvested at 24hpi. Western blots show time-dependent depletion of E, core, NS4B, and NS5 in the presence GNF-2-deg and 2-12-2-deg. Representative results are shown from $n = 2$ independent experiments.

Response Figure 2. (A) Schematic of virus-like particle (VLP) time of addition assay. **(B)** The compound is added into the transfected cell at 4, 7, 10, and 16 hours post-transfection to concentration of 5 μ M. The supernatant and cell lysates are collected for the western blot analysis at 28 hours post-transfection. Representative results are shown from $n = 2$ independent experiments.

To summarize the results in our original submission and the additional ones described here:

1. Treatment with GNF-2-deg and 2-12-2-deg starting at 1 hour post-infection results in depletion of E, core, NS4B, and NS5 when analyzed at 20, 22, and 24 hours post-infection.
2. Some of the observed depletion of the other viral proteins at these time points might stem from general antiviral activity due to reduced progeny virus and thus reduced second cycle of infection.
3. The time of addition studies indicate that after 11 hours post-infection, there is a pool of E that is not susceptible to targeted protein degradation induced by GNF-2-deg and 2-12-2-deg. This is consistent with the idea that the E degraders can target E in the context of the viral polyprotein; however, proving this will require significantly more work, which we hope the reviewer agrees goes beyond the scope of the study.
4. The loss of antiviral activity against the F193L/M196V double mutant in the VLP assay shows that the antiviral effects of GNF-2-deg and 2-12-2-deg are mediated by their interactions with E. This may be E that has been cleaved from the polyprotein or E and/or E that is still part of the polyprotein. Further experiments are required to resolve this. We thank the Reviewer for stimulating us to begin investigating this.
5. We cannot exclude the possibility that the GNF-2-deg and 2-12-2-deg affect other steps in the viral replication cycle. If the E degraders are targeting E within the context of the polyprotein, we would expect there to be antiviral effects on viral genome replication and polyprotein processing due to depletion of the nonstructural proteins required for these viral processes.

(2) Two minor points on Figure 2: Why was an MOI of one chosen since not every cell will be simultaneously infected? Why do the authors graph the X-axis with a decreasing concentration of PROTAC rather than increasing?

Response: We thank the reviewer for the comment. We have chosen MOI 1 as the experimental condition given the value implies an average probability of 1 infectious viral particle infecting one cell. This MOI is also consistent with our prior work characterizing the antiviral activities of the parental inhibitors GNF-2 and 2-12-2. For the Western blot analysis, we have the sample with compound treatment from left-to-right 20 μ M to 0 μ M with 2-fold dilution. However, the x-axis of viral titer results is shown with an increasing concentration of compounds. We apologize for any confusion regarding this.

(3) Figure 3: It is interesting that the authors chose to measure virus titer as % DMSO. I find this difficult to follow. What is the absolute reduction in infectious viral units of the PROTACs inhibition?

Response: We thank the reviewer for the comment. We have attached the data with the unit of pfu/mL. The absolute reduction in infectious viral units is in the range of 400000-600000 pfu/mL.

(4) Figure 4: It would be useful if the bands seen in Panels A and D were quantified for easier comparison.

Response: We thank the reviewer for this comment. We revised Figure 4 and included the quantitative analysis of the band in both Panel A and D.

(5) Line 441: Have the authors considered assessing whether particles are budding into the ER lumen and thus degradation of E comes at a later step?

Response: We agree with the Reviewer's comment. As described in our response to Comment 1, we conducted additional experiments to begin investigating this. In both the infectious virus and VLP systems, we observed that maximum depletion of E occurs when the compound treatment is added earlier than 11 hours post-infection or post-transfection. At 11 hours post-infection and later, we observe intracellular E that does not appear susceptible to degradation. This is consistent with the idea that E is no longer susceptible once it is on particles that have budded. Experiments to show definitively what form of E is being targeted by GNF-2-deg and 2-12-2-deg are a significant undertaking, which we have begun, but which we hope the Reviewer can agree go beyond the scope of the current study.

REVIEWER COMMENTS

Reviewer #2 (Remarks to the Author):

In this revised manuscript, the authors explore the use of proteolysis targeting chimeras (PROTACs) directed at the dengue virus E protein. This is an exciting new approach in which they demonstrate that their PROTACs have a dual antiviral activity. These are exciting and important new studies. The authors seriously addressed the concerns of the initial review. In particular was the question around specificity of protein degradation. Surprisingly they found after additional experiments that several of the dengue proteins were degraded following treatment. They suggested two hypotheses but are unable to narrow it down at this point. This will be important to determine but perhaps not necessary for this manuscript. However, the authors should point this observation out in the manuscript.

Reviewer #2 (Remarks to the Author):

In this revised manuscript, the authors explore the use of proteolysis targeting chimeras (PROTACs) directed at the dengue virus E protein. This is an exciting new approach in which they demonstrate that their PROTACs have a dual antiviral activity. These are exciting and important new studies. The authors seriously addressed the concerns of the initial review. In particular was the question around specificity of protein degradation. Surprisingly they found after additional experiments that several of the dengue proteins were degraded following treatment. They suggested two hypotheses but are unable to narrow it down at this point. This will be important to determine but perhaps not necessary for this manuscript. However, the authors should point this observation out in the manuscript.

Response:

We thank the Reviewer for the excellent suggestions during the review process. We want to note that we had added text to the Discussion in our last revision, mentioning the effects on core, NS4B, and NS5, but we neglected to highlight this in our Response to Reviewers. We had also not shown the data because we conservatively felt reluctant to report the data without having done more experiments to enable accurate interpretation of the data. In this second revision, we have modified the manuscript as requested to point out the observations we made regarding depletion of the other viral proteins. The changes are as follows:

1. We have added the Western blot data for core, NS4B, and NS5 as **Supplementary Figure 4** and added the following text to the Results section on page 12:

“Western blot analysis of core, NS4B, and NS5 in cell lysates indicated that these viral proteins were also depleted in the presence of GNF-2-deg and 2-12-2-deg at this time point and at the earlier time points at which we could reproducibly detect these proteins under these infection conditions (**Supplementary Figure 4**). The reduced abundance of core, NS4B, and NS5 in these end-point measurements may reflect reduced expression of these proteins due to general antiviral activity and/or increased degradation of core, NS4B, and NS5 in the presence of GNF-2-deg and 2-12-2-deg, perhaps due to effects of the compounds on the viral polyprotein (**Supplementary Note 1**). With respect to mode of action, these observations opened the possibility that reduced viral gene expression and/or reduced RNA replication could contribute to the antiviral activity observed in these experiments.”

2. We added **Supplementary Note 1** to provide a fuller discussion of why we believe that the depletion of core, NS4B, and NS5 is not due to general antiviral activity.

3. We added the data from the live virus and VLP experiments showing that we do not observe significant depletion of E if the compound is added at 12 hours post-infection (virus) or 10 hours post-transfection (VLP) as **Supplementary Figure 6**. This is now accompanied by the following text on pages 20-21 of the revised manuscript:

“Assuming that the effects of the two E PROTACs are ‘on-target,’ there are two possibilities to consider. First, small molecules may bind to E on the luminal side of the ER membrane, with E subsequently relocalized (via an uncharacterized mechanism) to the cytosol where ternary complex formation and/or proteasomal degradation of E can occur. The observation of a pool of E that is not susceptible to degradation when GNF-2-deg and 2-12-2-deg are added at 16 hours post-transfection in the VLP system or 12 hours post-infection in the live virus system (**Supplementary Figure 6**) argues against this, as it suggests that E may no longer be susceptible to these PROTACs once it is on a viral particle that has entered the secretory system. A second hypothesis is that GNF-2-deg and 2-12-2-deg engage E prior to its insertion

through the ER membrane, with ternary complex formation occurring while the protein is still on the cytosolic side of the ER membrane. In this case, the actual target may be the viral polyprotein, and small molecule-binding may potentially delay translocation across the membrane. The disappearance of other viral proteins (core, NS4B, NS5) in the presence of the E PROTACs (**Supplementary Figure 4**) is consistent with this hypothesis although further experiments are needed to demonstrate polyprotein targeting explicitly and to rule out reduced viral gene expression due to general antiviral activity.”